# Genomic Landscape of Mixed-Phenotype Acute Leukemia

**DOI:** 10.3390/ijms231911259

**Published:** 2022-09-24

**Authors:** Marah Hennawi, Nagehan Pakasticali, Hammad Tashkandi, Mohammad Hussaini

**Affiliations:** Department of Pathology and Laboratory Medicine, Moffitt Cancer Center, Tampa, FL 33612, USA

**Keywords:** mixed-phenotype acute leukemia, mutations, NGS, genomic, targeted therapy

## Abstract

Mixed-phenotype leukemia (MPAL) is a type of acute leukemia in which the blast population shows mixed features of myeloid, T-lymphoid, and/or B-lymphoid differentiation. MPALs are rare and carry a poor prognosis, thus, often pose both a diagnostic and therapeutic challenge. Conventionally, the diagnosis of MPAL requires either a single blast population with a lineage-defining phenotypic expression of multiple lineages (myeloid, B-cell and/or T-cell) (biphenotypic) or two distinct blast populations that each independently satisfy criteria for designation as AML, B-ALL, and/or T-ALL (bilineage). Given the rarity of MPAL, minimal studies have been performed to describe the genomic landscape of these neoplasms. IRB approval was obtained. Central MCC database was searched for any patient with a diagnosis of acute undifferentiated leukemia (AUL), acute leukemia of ambiguous lineage (ALAL), and MPAL. All patient diagnoses were manually reviewed by a hematopathologist to confirm the diagnosis of MPAL. Genomic and molecular data were collated from the EMR and bioinformatically from MCC genomics repositories. Twenty-eight patients with MPAL were identified. Thirteen were female and 15 were male. Average age was 56 years old (range = 28–81). Ten cases were biclonal and 18 were biphenotypic. Diagnoses were as follows: B/myeloid (*n* = 18), T/myeloid (*n* = 9), and T/B (*n* = 1). Cytogenetic analysis (Karyotype +/− FISH) was available for 27 patients. The most frequent recurrent abnormalities were complex karyotype (*n* = 8), BCR/ABL1 translocation (*n* = 6), Del 5q/−5 (*n* = 4), Polysomy 21 (*n* = 4). Mutational analysis was available for 18 patients wherein mutations were detected in 45 unique genes. The most frequently mutated genes were *TP53* (7), *RUNX1* (6), *WT1* (4), *MLL2* (3), *FLT3* (3), *CBL* (2), *ASXL1* (2), *TET2* (2), *MAP3K6* (2), *MLL* (2), and *MAP3K1* (2). Targetable or potentially targetable biomarkers were found in 56% of cases. Overall survival was 19.5 months (range = 0–70 m). Ten patients were treated with an allogeneic stem cell transplant and had superior outcome (*p* = 0.0013). In one the largest series of MPAL cases to date, we corroborate previous findings with enriched detection of RUNX1 and FLT3–ITD mutations along with discovery of unreported mutations (*MAP3K*) that may be amenable to therapeutic manipulation. We also report the frequent occurrence of AML with MDS-related changes (AML-MRC)-defining cytogenetic abnormalities (26%). Finally, we show that those patients that received stem cell transplant had a better overall survival. Our findings support the need to genomically profile MPAL cases to exploit opportunities for targeted therapies in this orphan disease with dismal prognosis.

## 1. Introduction

Mixed-phenotype leukemia (MPAL) is a type of acute leukemia in which the blast population(s) show mixed features of myeloid, T-lymphoid, and/or B-lymphoid lineage [1]. MPALs are rare, comprising only 2–5% of acute leukemias, and carry a poor prognosis, thus, often posing both a diagnostic and therapeutic challenge [2]. The first consensus algorithm for diagnosing MPAL was proposed by the European Group for Immunological Characterization of Acute Leukemias (EGIL) in 1995 and involved a point system based on expression of various phenotypic markers by the blasts. This classification gave way to the World Health Organization (WHO) criteria which require either a single blast population with lineage-defining phenotypic expression of multiple lineages (myeloid, B-cell and/or T-cell) (biphenotypic) or two distinct blast populations that each independently satisfy criteria for designation as AML, B-ALL, and/or T-ALL (bilineage). In this scheme, emphasis is placed on certain lineage-defining markers, namely CD19, CD3 (cytoplasmic or surface), and myeloperoxidase for B-cell, T-cell, and myeloid lineages, respectively [3]. The new proposed International Consensus Classification (ICC) classification incorporates *ZNF384* rearrangement and myeloid/lymphoid neoplasm with eosinophilia (M/LN-eo) with tyrosine kinase (TK) gene fusions which can present as B/myeloid MPAL. However, the immunophenotypic criteria for MPAL do not appear to be modified [4]. Additionally, the newest rendition (5th edition) of the WHO Classification of hematolymphoid tumors categorizes acute leukemias with *ZNF384* rearrangement or *BCL11B* rearrangement under acute leukemia of ambiguous lineage with defining genetic abnormalities, even if they qualify as MPAL phenotypically. The criteria for immunophenotypically defined MPAL remains largely unchanged according to the WHO criteria, although antigen expression parameters have been revised and refined [5]. Overall, given the rarity of MPAL, there are only a few studies describing the genomic landscape of these neoplasms. This is important since these primitive neoplasms have overlapping clinical, immunophenotypic, and genetic features, suggesting a biological spectrum amongst acute leukemias warranting additional genomic data to guide classification, and subsequently, patient management. In particular, genetic profiling for personalized treatment and prognosis makes additional genomic characterization necessary. In this study, we report on one of the largest cohorts of MPAL in the literature, along with their genomic and clinical characteristics.

## 2. Results

Twenty-eight patients with MPAL were identified. Thirteen were female and 15 were male. The average age was 56 years old (range= 28–81 years). Ten cases were biclonal and 18 were biphenotypic. Diagnoses were as follows: B/myeloid (*n*= 18), T/myeloid (*n* = 10), and T/B (*n* = 1).

Cytogenetic analysis (Karyotype and or FISH) was available for 27 patients (Table 1). The most frequent recurrent abnormalities were complex karyotype (8), BCR/ABL1 translocation (6), Del 5q/−5 (4), and Polysomy 21 (4). In our cohort, additional MDS-related cytogenetic abnormalities included del 7/−7 (3), Trisomy 8 (1), Del 17p/−17 (2), del 20q (1), del 9q (1), and del 13q (1). No patients with *ZNF384* rearrangements were found.

There were three patients with rearrangements involving 14q23, which may result in the dysregulation of *BCL11B* [6]. In two patients, karyotype showed 45, XY,t(2;14;5)(q23;q32;q13), der (12;16) (q10;p10), add (17)(q21) and 46,XX,t(2;14)(p13;q32)?c [20]. ISH on the lymph node of the third patient identified a karyotypically occult translocation (5;14) resulting in the TLX3–BCL11b fusion (74%). All three showed T/myeloid disease. NGS showed *WT1* C265R (VAF: 49%) in the second patient and *WT1* (81%) and *PHF6* (40%) alterations in the third patient’s lymph node. NGS was not performed on the first patient.

For those who were *BCR/ABL1*-positive (6), three were p210-positive and two were p190-positive, and one patient was positive for both p210 and p190.

Four patients had *TP53* mutations, two of which had complex karyotype.

Other rearrangements included t(1;16) (q12;q11.2), which has been previously described in B-lymphoid blast crisis of chronic myeloid leukemia. Other novel translocations in MPAL included t(1;3) (q21;p25). An identical translocation involving *BCL9* has been previously reported in Burkitt lymphoma [7]. Finally, the canonical t(8;14)(q24;q32) of Burkitt lymphoma was detected in one case. This is the first report of this translocation in MPAL. The patient had a history of HIV and hepatitis C and presented diplopia, B symptoms, abdominal pain and CSF involvement. He expired within a week of presentation. Cytogenetic data are summarized in Table 1 and Figure 1.

Mutational analysis was available for 18 patients wherein mutations were detected in 45 unique genes. The most frequently mutated genes were *TP53* (7), *RUNX1* (6), *WT1* (4), *MLL2* (3), *FLT3* (3), *CBL* (2), *ASXL1* (2), *TET2* (2), *MAP3K6* (2), *MLL* (2), and *MAP3K1* (2) (Figure 1 and Figure 2). Mutational data are summarized in Figure 2 and Figure 3.

There was considerable heterogeneity in treatment. Most patients received ALL therapy with most receiving Hyper-CVAD, or modifications thereof (B/myeloid = 12; T/myeloid = 2). Some patients received myeloid therapy such as hypomethylating agent (HMA) (T/myeloid = 1, B/myeloid = 1 (+Venatoclax)) or 7 + 3 (B/myeloid = 3, T/myeloid = 2). Relapses were treated with Midastaurin, HMA, CLAG-M, Sorefenib, or Venetoclax + HMA.

A subset of patients were found to have targetable mutations. Of the three patients with *FLT3*–ITD mutations, one (B/Myeloid) was treated with adjuvant Midastaurin initially and adjuvant Gilteritinib at relapse, with a response. The other patient (T/myeloid) received adjuvant Sorefanib with HyperCVAD for reinduction upon relapse and achieved complete remission, but later relapsed. The last patient was diagnosed with MPAL prior to the release of *FLT3*-targeted therapy. One patient with targetable *IDH2* R140Q was detected but he expired in 14 days. We also note the recurrent involvement (*n* = 3) of the MAPKKK family of serine/threonine-specific kinases involved in proliferation and differentiation, which warrants an additional investigation as a target. Lastly, the *KMT2* (*MLL*) family was also noted to be recurrently involved (*n* = 3), raising the possibility of targeting with anti-KMT2A (KO-539), of which is in trial.

Overall survival was 19.5 months (range= 0–70). Ten patients were treated with an allogeneic stem cell transplant. Mean survival for those with transplant was 34.1 months (SD 21.5) versus 11.4 months (SD 11.9) for those without (*p*= 0.0013).

## 3. Discussion

Early studies reporting on MPAL either lacked mutational data, only looked at limited genes, or involved the analysis of one to two patients [8,9,10,11]. A notable exception is what is still one of the largest series of MPAL (*n* = 117) to date, with diseases classified based on the 2008 WHO criteria. In that study, cytogenetic data were available for 92 patients; in order of frequency, they showed complex karyotype (24%), *BCR/ABL1* fusion (15%), −7 (8%), Polysomy 21 (8%), and t(v;11q23) (4%). They performed a mutational analysis on 31 patients for 17 genes and found mutations in 12 patients—the most common being: *IKZF1* deletions (13%), *EZH2* (9.7%), and *ASXL1* (6.5%). No mutations in *DNMT3A, FBXW7*, *FLT3*, *IDH1*, *IDH2*, *KIT*, *NPM1*, *PHF6*, *RUNX1*, or *WT1* were reported [12].

Subsequently a large multicenter US-based study identified nine cases of B/T MPAL, where recurrent mutations in *PHF6* and the involvement of JAK-STAT and Ras signaling pathways were reported [13].

In a more recent study by Takahashi et al., an integrated genomic analysis was performed on 31 MPAL samples. Abnormal karyotype was found in nearly 70% of cases. Overall, 26% had complex karyotype abnormalities, 13% were Philadelphia chromosome (Ph+)-positive, and one case had a 11q23 rearrangement (t [11;19][q23;p13.3]). They examined 295 genes in all 31 patients and found mutations in 94% of cases. The most common mutations were found in *NOTCH1* (29%), *RUNX1* (26%), and *DNMT3A* and *IDH2* (23% each) [14]. B/myeloid cases were enriched for *RUNX1* mutations. In our data set, this was confirmed, as *RUNX1* mutations were seen exclusively with the B/myeloid phenotype.

Limited studies have investigated pediatric acute leukemia of ambiguous lineage (ALAL). A large multicenter, international study looked at 110 cases of MPAL and found 81 recurrently mutated genes to include *FLT3* (27%), *RUNX1* (13%), *CDKN2A* or *CDKN2B* (19%), *ETV6* (20%), *VPREB1* (13%), *WT1* (24%), and *KMT2A* (23%) [15]. However, genes involved in adult MPAL (e.g., *IDH2*, *DNMT3A*) appeared to be absent in the pediatric MPAL [16]. These data are summarized in Table 2.

In our study, we confirm many of the previous findings but expand on the data in significant ways. In one of the largest series of adult MPAL cases with cytogenetic and mutational results, first we found the frequent occurrence of AML-MRC-defining cytogenetics abnormalities (*n* = 7, 26%) and MDS-defining chromosomal abnormalities (*n* = 12; 44%), even after excluding cases with complex karyotype. There is ambiguity in the 2016 WHO classification with regard to how to categorize these cases. On the one hand, it has been suggested that cases of AML with MDS-related changes (AML-MRC) should be designated as such with a note regarding mixed-phenotype of blasts. However, MDS-defining cytogenetic abnormalities are also listed in the genetic profile of neoplasms categorized as B/myeloid MPAL by 2008 WHO. To address this conundrum, we examined the clinical behavior of these groups by comparing the OS of MPAL patients with and without AML-MRC-defining cytogenetic abnormalities. Those who harbored these abnormalities had a lower overall survival (11.14 months) versus those that did not (22.95 months) (*p* = 0.1697). Larger cohort studies will be needed to further refine diagnostic criteria and confirm these findings.

Next, we looked at those patients with an exceptionally poor prognosis (OS less than 4 months; *n* = 8). Blast percentage was high at diagnosis (86%). They were B/myeloid (6), T/myeloid (1) and T/B (1). One half had complex karyotype. No particular set of mutations was enriched in this group. Treatment strategies were heterogenous like the overall group, with the exception that only one of these patients received a transplant.

Next, we reported a different set and distribution of genetic alterations consistent with long-tail distribution of mutations in lymphomas [17]. Forty-five unique genes were found to be mutated, more than reported in prior studies. The most frequently mutated genes included were *TP53* (7), *RUNX1* (6), and *WT1* (4). In addition, we found a substantial subset of patients with targetable or potentially targetable disease (56% of those who were profiled). Three patients had *FLT3*–ITD, two of whom received TKI therapy with a response. Another patient had *IDH2* R140Q, which would have been amenable to enasidenib therapy had the patient survived [18]. Three patients had aberrations of the KMT2 family, thus, providing a basis for an investigation on anti-KMT2A therapy. KO-539 is currently in clinical trial (NCT04067336) and preliminary data have shown anti-leukemic activity in patient-derived xenograft (PDX) models [19]. Uniquely, we discovered a recurrent involvement (*n* = 3) of the MAPKKK family in serine/threonine-specific kinases, of which are involved in cellular proliferation and differentiation. Several FDA-approved therapies already target MAPK signaling, such as BRAF and MEK inhibitors, and their use in MPAL patients with alterations in this pathway is worthy of further investigation.

Twenty patients achieved complete remission (CR), while seven patients demonstrated persistent disease. The overall survival in the group with persistent disease was dismal (2.9 months). In the CR group, the mutational profile included *TP53* (4), *RUNX1* (3), and *FLT3* (2). Cytogenetics showed del 5q (2), complex (3), and t(9;22) (2) in this group.

Finally, we confirm the dismal prognosis of patients with MPAL diagnosis (19.5 months). We note a wide variance in treatment strategies but show that hematopoietic stem cell transplantation is an effective treatment for these patients. The mean survival for those with a transplant was significantly superior at 34.1 months (SD 21.5) versus 11.4 months (SD 11.9) for those without a transplant (*p* = 0.0013).

## 4. Materials and Methods

IRB approval was obtained. The Central Moffitt Cancer Center (MCC) database was searched for any patients with a diagnosis of acute undifferentiated leukemia (AUL), acute leukemia of ambiguous lineage (ALAL), or mixed-phenotype acute leukemia (MPAL) from a database of >600,000 entries. All patient diagnoses were manually reviewed by a board-certified hematopathologist to confirm the diagnosis based on the 2016 WHO classification. Cases with myeloid component and complex karyotype were classified as MPAL and were included in the cohort. Genomic and molecular data were collated from the electronic medical record and bioinformatically from MCC genomics repositories. Next generation sequencing data were derived from either Foundation One Heme^®^ Testing or in-house CAP/CLIA-certified Illumina-based myeloid NGS panels, or 98-gene myeloid (Sophia-based custom Myeloid Action Panel) NGS panels performed on these patients.

FoundationOne Heme (Foundation Medicine, Cambridge, Massachusetts) is a combined DNA- and RNA-sequencing assay that uses hybrid-capture NGS technology to detect single nucleotide base substitutions, insertions and deletions, copy number variants, and certain fusions across >400 protein-coding genes, 250 RNA transcripts, and 30 selected introns that are potentially involved in hematologic malignancies. Sensitivity, specificity, and reproducibility were reported to be greater than 95% [20,21].

NGS was performed at the Moffitt Cancer Center in a CAP/CLIA-certified environment using a custom TruSeq myeloid 32-gene panel that was transitioned to an Illumina TruSight Myeloid 54-gene panel in 2016 [22]. Insertions/deletions were reported at a validated variant allele frequency (VAF) of >10%. Single nucleotide variants were reported with a variant allele frequency (VAF) of ≥ 5% in all tests.

The Moffitt-MAP assay is based on a customized NGS assay from SOPHiA Genetics and Illumina sequencing technology platform. For library generation, a hybridization-capture-based enrichment method was employed; Illumina sequencing by synthesis (SBS^®^) technology was used to interrogate the following genes: *ABL1*, *ANKRD26*, *ASXL1*, *ASXL2*, *ATM*, *ATRX*, *BCOR*, *BCORL1*, *BRAF*, *BRCC3*, *CALR*, *CBL*, *CBLB*, *CBLC, CCND2, CDKN2A, CEBPA, CHEK2, CREBBP, CSF3R, CSMD1*, *CSNK1A1*, *CTCF*, *CUX1*, *DDX41*, *DHX15*, *DNMT3A*, *ELANE*, *ETNK1*, *ETV6*, *EZH2*, *FANCA*, *FANCL*, *FLT3*, *GATA1*, *GATA2*, *GNAS*, *GNB1*, *HNRNPK*, *HRAS*, *IDH1*, *IDH2*, *IKZF1*, *JAK1*, *JAK2*, *JAK3*, *KDM6A*, *KIT*, *KMT2A*, *KMT2D*, *KRAS*, *LUC7L2*, *MECOM*, *MET*, *MPL*, *MYC*, *NF1*, *NOTCH1*, *NOTCH2*, *NPM1*, *NRAS*, *PAX5*, *PDGFRA*, *PHF6*, *PIGA*, *PML*, *PPM1D*, *PTPN11*, *RAD21*, *RAF1*, *RB1*, *RBBP6*, *RPS19*, *RTEL1*, *RUNX1*, *SAMD9*, *SAMD9L*, *SBDS*, *SETBP1*, *SF3B1*, *SH2B3*, *SMC1A*, *SMC3*, *SOS1*, *SRP72*, *SRSF2*, *STAG1*, *STAG2*, *STAT3*, *STAT5B*, *TERC*, *TERT*, *TET2*, *TP53*, *U2AF1*, *WT1*, *ZBTB7A*, *ZRSR2.* Insertions/deletions, as well as single nucleotide variants, were reported at a variant allele frequency cut-off >3%.

## 5. Conclusions

In summary, our findings expand on the molecular underpinnings of MPAL and may carry prognostic implications in a disease subset with already a dismal prognosis. In one of the largest series of MPAL cases to date, we corroborate previous findings with an enriched detection of *RUNX1* mutations, along with the discovery of unreported mutations (*MAP3K*) that may be amenable to therapeutic manipulation. Our findings support the need to genomically profile MPAL cases to exploit opportunities for targeted therapies in this orphan disease with dismal prognosis. In addition, we note the frequent occurrence of AML-MRC-related cytogenetic abnormalities in MPAL patients which seem to carry a worse prognosis and may qualify for their own designation as a disease category; however, larger corroborative studies are needed. Lastly, we provide evidence for stem cell transplants as an effective treatment strategy for patients with this aggressive disease.

## Figures and Tables

**Figure 1 ijms-23-11259-f001:**
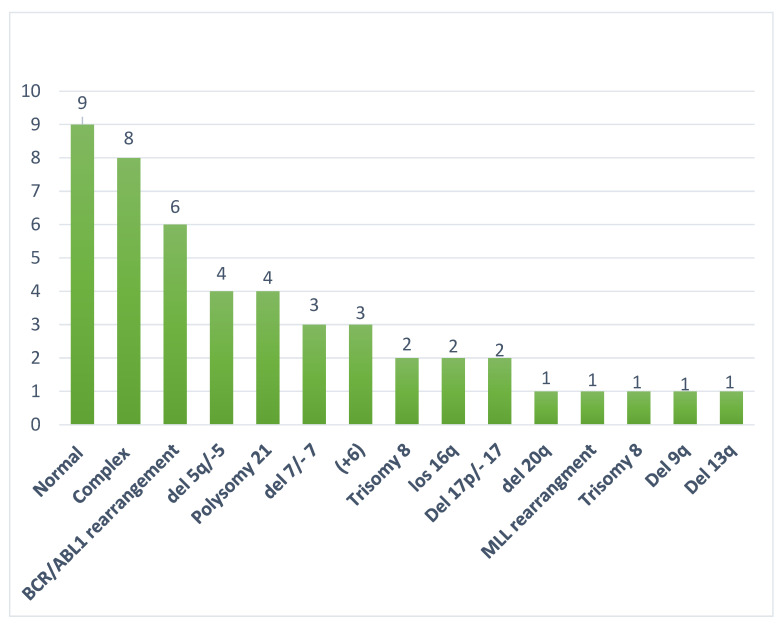
Frequency of common cytogenetic alterations in mixed-phenotype acute leukemia (MPAL).

**Figure 2 ijms-23-11259-f002:**
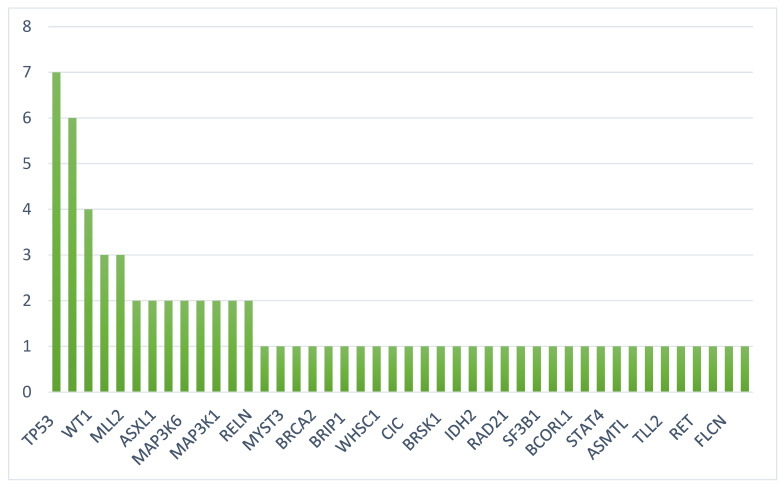
Mutation frequency in mixed-phenotype acute leukemia (MPAL).

**Figure 3 ijms-23-11259-f003:**
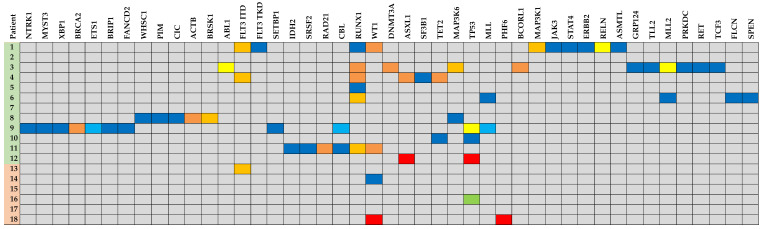
Genomic landscape of mixed-phenotype acute leukemia (MPAL).

**Table 1 ijms-23-11259-t001:** Summary of karyotypic and fluorescence in situ hybridization (FISH) findings.

Diagnosis	Clonality	Cytogenetics	FISH
**B/myeloid**	Biclonal	47,XX,+6 [3]/46,XX [17]	BCR/ABL1-negative, MDS FISH-negative.
**T/B lymphoid**	Biphenotypic	65, XY, +X, +1, 1, add(1)(p33)x2, −2, +3, −4, +5, del(5q31q35)x2, +6, +6, +7, +8, −9, +11, +20, +20, −20, +21, +22, +mar [1]/46, XY [2]	
**T/myeloid**	biphenotypic	45,XY,t(2;14;5)(q23;q32;q13), der (12;16) (q10;p10), add (17)(q21)	Monosomy 16/16q, No rearrangement of CBFB, MLL, BCR/ABL, RUNX1T1/RUNX1, or PML/RARA.
**B/myeloid**	Biphenotypic	46,XY [20]	Normal for MLL, 5q and 7q.
**T/myeloid**	Biphenotypic	47,X,add(X)(p22.1),−5,add(6)(q13),−7,add(9)(p11),del(9)(q13q22),+21,+2mar [6]/ 46,XX [17]	No evidence of a BCR–ABL rearrangement.
**T/myeloid**	Biphenotypic	46,XX,t(2;14)(p13;q32)?c [20]	PML/RARα rearrangement is not detected.
**B/myeloid**	Biclonal	t(9;22)(q34;q11.2);46, XY, der(16)t(1;16)(q12;q11.2),t(9;22)(q34;q11.2) [19]/46, XY [1].	t(9,22) along with loss of CBFB/16q, and gain in RUNX1/21q.
**B/myeloid**	Biphenotypic	No data available	
**B/myeloid**	Biphenotypic	47,XY,t(2;17;8)(p23;q25;q22),t(9;22)(q34;q11.2),del(13)(q22q32),+21 [cp20]	t(9;22)(q34;q11.2) translocation (97.5%) and gain of extra Ph chromosome.
**B/myeloid**	Biclonal	46,XX [20]	No evidence of a BCR/ABL1 gene rearrangement; deletion of 5q, 7q, 17q, and 20q not detected. Monosomy 7 and trisomy 8 not detected.
**B/myeloid**	Biclonal	46,XX,?ins(1;1)(p34;p32p36.1),t(9;22)(q34;q11.2) [18]/92,idem x2 [2]	A total of 5.5% of cells with tetrasomy 8 and tetrasomy 21. No evidence of RUNX1T1/RUNX1; No MLL gene rearrangement. The 4 MLL probe fusion signals (13.5%) indicate the presence of tetraploidy tumor clone in the specimen.
**B/myeloid**	Biphenotypic	46,XY,t(9;22)(q34;q11.2) [4]	Normal CDKN2A(P16), ETV6/RUNX1, TCF3; t(9;22)(q34;q11.2)(Normal).
**B/myeloid**	Biclonal	46,XX,t(4;11)(q21;q23) [10]/46,XX [1]	MLL rearrangement (12%).
**T/myeloid**	Biphenotypic	46,XY [20]	Deletion of 5q, 7q, 17p, and 20q not detected. Monosomy 7 and trisomy 8 not detected.
**B/myeloid**	Biphenotypic	45,XY,−7,t(9;22)(q34;q11.2) [15]/46,XY [5]	BCR/ABL1 rearrangement (92.5%). Positive for deletion of 7q or monosomy 7 (89%).
**B/myeloid**	Biclonal	46,XX [20]	No BCR/ABL1.
**B/myeloid**	Biclonal	93–108,XXYY,−2,−3,−6,−6,−7,−8,−8,−9,add(11)(q23)x2, −12,−12,+15,+10-25mar[cp3]/46,XY [19]; POSSIBLE KMT2A (MLL) ONCOGENE MEDIATED CLONE DETECTED	No BCR–ABL1, MLL or RUNX1/RUNXT1 rearrangements. Increased ABL1 and BCR signals, evidence of MLL gene amplification and increased RUNX1, RUNXT1 signals.
**B/myeloid**	Biphenotypic	46,XX [20]	Positive for del(17p/TP53), loss of ABL1 and BCL-6 genes, gain of BCR gene, and negative for BCR/ABL1 fusion, t(11;14), or BCL-2/BCL-6/MYC rearrangements.
**B/myeloid**	Biphenotypic	46,XY,+13,−21 [20]	Negative for PML–RARA, RUNX1–RUNXT1, CBFB–MYH11, BCR–ABL1, and MLL/KMT2A translocations. MDS FISH panel is normal.
**T/myeloid**	Biphenotypic	39–46,XX,-X,add(X)(p22.1),add(7)(q11.2),add(11)(p11.2),−14,−16,−17,−18,−20,del(20)(q11.2q13.3),add(21)(q22),−22,add(22)(q11.2)+r,+1-3mar[cp20]	Deletion 5q31-negative, deletion 7q31-positive, deletion 20q12-positive, deletion 17p13 (p53)-negative, Trisomy 8-positive.
**B/myeloid**	Biphenotypic	46,XX [20]	No RUNX1T1, RUNX1 (ETO/AML1), MLL GENE, BCR/ABL1-negative; Del 5q31 detected at relapse.
**T/myeloid**	Biphenotypic	46,XX,del(5)(q22:q35),del(11)(p11.2) with an abnormal BCR–ABL1 signal with only one ABL1 gene at 9q34 detected; First relapse: 48,XX,add(1)(p36.3),del(5)(q22),+6, del(11)(p11.2),+19[cp17]/46,XX [3]	FISH negative for BCR/ABL1; no AML OR ALL gene deletions or rearrangements.
**B/myeloid**	Biphenotypic	45,XX,der(3)t(1;3)(q21;p25), t(8;14)(q24;q32),der(14)t(8;14),−15, del(17)(p11.2),add(19)((q13.4)[cp20]	MLL deletion (63%); no BCR–ABL1 fusion.
**B/myeloid**	Biclonal	46,XY [20]	No t(8;21), t(9;22), 11q23, t(15;17) or inv16.
**T/myeloid**	Biphenotypic	46,XY,dup(1)(q23q32),del(3)(q21), +13,−21[cp10]/46,XY[4]	MDS FISH panel is normal; no t(9;22) translocation.
**T/myeloid**	Biphenotypic	46,XY [20]	No PML/RARA gene rearrangement.
**T/myeloid**	Biclonal	46, XX	Karyotypically occult translocation (5;14) resulting in the TLX3–BCL11b fusion (74%) with 60% heterozygous/hemizygous deletion of CDKN2a and with 8.5% of nuclei demonstrating a TRB rearrangement involving 7q34.
**B/myeloid**	Biclonal	46,XX,add(5)(q11.2),t(9;22)(q34.1;q11.2) [18]/47,idem,+der(22)t(9;22)[1]/46,XX [1].	Positive for t(9;22), and negative for t(1;19), MYC, KMT2A or IGH gene rearrangements, negative for Trisomy 4, 6, 10 or 17.

**Table 2 ijms-23-11259-t002:** Summary of prior genomic studies of MPAL.

		Heesch, S. et al. [8]	Eckstein, O.S. et al. [9]	Quesada, A.E. et al. [10]	Matutes, E. et al. [11]	Yan, L. et al. [12]	Mi, X. et al. [13]	Takahashi, K. et al. [14]	Alexander, T.B. et al. [15]	Becker, M. et al. [16]
**Sex**										
	Female	21			38		57	13		
	Male	21			62		60	18		
**Age, years**										
	Median	60	35				35	53		7
**Diagnosis**										
	AUL	16			13		26		5	
	ALL				39		51			
	AML				38		40			
	MPAL (B/Myeloid)	12	7	7	59		64	13	35	37
	MPAL (T/Myeloid)	12	15	6	35		38	18	49	52
	MPAL (B/T-cell	2	1	1	4		14			
	MPAL (B/T/Myeloid)				2		1			
**Cytogenetics**										
	Normal Karyotype	3	4			10	33		5	
	Complex karyotype					24	22	8		
	t(9;22)(q34;q11)	7	1	1		15	14	4		2
	Monosomy		5				7			
	Polysomy						12			
	t(v;11q23)					6	4	1		
	t(10;11)(p15;q21)						3			
	MLL		2			7				15
	Other abnormalities		11			21		21		
**Mutations**										
	*WT1*	4	3	3			0	1	28	24
	*FLT3*	1	3	3			0		31	21
	*DNMT3A*		6	1			0	7		
	*MLL*		2					1		
	*RUNX1*		4	2		8	1	8	15	13
	*IDH2*		2	1			0			
	*TP53*		5	1				2		
	*JAK2*		1	1					1	
	*NOTCH1*		5	1			1	9		
	*NRAS*		4	1				6	21	18

## Data Availability

Data will be made available upon reasonable request to the corresponding author.

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
