# Peer review of "Genomic Landscape of Mixed-Phenotype Acute Leukemia"

_ijms, 2022, doi:10.3390/ijms231911259_

Round 1

Reviewer 1 Report

Mixed phenotype leukemias (MPALs) are a rare subtype of acute leukemia.   Leukemic blasts from MPAL patients display a mixed feature of both myeloid and lymphoid lineages.  Patients with such diseases are often challenge diagnostically and therapeutically with poor prognosis.  Due to the small number of cases have been reported, the genomic landscape of this subtype of leukemia is less studied.  In this study, by searching the database at Central Moffitt Cancer Center (MCC), Dr. Hennawi et al., identified 29 MPAL cases that had been treated in MCC.  By analyzing the genetic mutation  profile and clinical features of the patients, the authors reported the genetic mutations identified in these cases.  The authors also discussed the correlation of these mutations with patient outcomes and indicated some of the potential therapeutic targets.   The manuscript is well-organized and well-written.  The experimental design is logical and reasonable.  The experimental result is solid. 

Major concern:

The data from several other previous studies are included in the discussion, however, such information is difficult to follow and cannot reach a logical conclusion.  It will be better if the authors can summarize the data from all these studies into a table.   

Reviewer 2 Report

I would ask if there are any informations about the outcome of patients who achieved CR and the morphological and genomic features of patients relapsed 
